# Left Atrial Appendage Amputation for Atrial Fibrillation during Aortic Valve Replacement

**DOI:** 10.3390/jcm11123408

**Published:** 2022-06-14

**Authors:** Jurij M. Kalisnik, Giuseppe Santarpino, Andrea I. Balbierer, Janez Zibert, Ferdinand A. Vogt, Matthias Fittkau, Theodor Fischlein

**Affiliations:** 1Department of Cardiac Surgery, Klinikum Nürnberg, Paracelsus Medical University Nuremberg, 90471 Nuremberg, Germany; ferdinand@vogt.at (F.A.V.); mfittkau@gmail.com (M.F.); tfischlein@gmail.com (T.F.); 2Faculty of Medicine, University of Ljubljana, 1000 Ljubljana, Slovenia; 3Klinikum Nürnberg, Paracelsus Medical University, Campus Nuremberg, 90419 Nuremberg, Germany; santarpino@unicz.it (G.S.); aibalbierer@gmail.com (A.I.B.); 4Department of Cardiac Surgery, Città di Lecce Hospital, GVM Care & Research, 73100 Lecce, Italy; 5Department of Experimental and Clinical Medicine, Magna Graecia University, 88100 Catanzaro, Italy; 6Department of Biostatistics, Faculty of Health Sciences, University of Ljubljana, 1000 Ljubljana, Slovenia; jzibert@gmail.com; 7Department of Cardiac Surgery, Artemed Clinic Munich-South, 81379 Munich, Germany

**Keywords:** ischemic stroke, atrial fibrillation, left atrial amputation, aortic valve replacement

## Abstract

Background. Occluding the left atrial appendage (LAA) during cardiac surgery reduces the risk of ischemic stroke; nonetheless, it is currently only softly recommended with “may be considered” by the current guidelines. We aimed to assess thromboembolic risk after LAA amputation in patients with atrial fibrillation (AF) and aortic stenosis undergoing biological aortic valve replacement (AVR) as primary cardiac surgery. Methods. Two cohorts were generated retrospectively: patients with AF undergoing AVR alone or combined with revascularization either with LAA amputation or without. Data were collected from the hospital-specific data system. Follow-up was completed by telephone interview or in person. Thirty-day and follow-up results were compared in patients with vs. without LAA amputation. Results. One hundred and fifty-seven patients were investigated retrospectively, and seventy-four pairs were matched with regard to baseline characteristics. Patients with LAA amputation exhibited a lower incidence of cumulative and late ischemic stroke (6.4% vs. 25%, *p* = 0.028 and 3.2% vs. 20%, *p* = 0.008, respectively; hazard ratio 0.30; 95% confidence interval 0.11; 0.84; *p* = 0.021) during follow-up of 48 months vs. patients without intervention during follow-up of 45 months, *p* = 0.494. No significant differences were observed in postoperative stroke, 2 (2.7%) vs. 3 (4.1%), *p* = 1.000, re-exploration for bleeding 3 (4.1%) vs. 6 (8.1), *p* = 0.494 or late pericardial effusion 2 (2.7%) vs. 3 (4.1%), *p* = 1.000, in-hospital 2 (2.7%) vs. 4 (5.4%), *p* = 0.681 and all-cause mortality 15 (23.8%) vs. 9 (15%), *p* = 0.315 in patients with vs. without LAA amputation, respectively. Conclusions. A combination of leading aortic stenosis and AF in patients undergoing isolated or combined biological AVR represents a subpopulation with excessive thromboembolic risk. Concomitant LAA amputation during cardiac surgery reduces the risk of ischemic stroke without posing an additional periprocedural risk for the patient. Therefore, the minimal invasive approach at the expense of omitting LAA amputation should be discouraged to maximize the clinical benefits of AVR in this setting.

## 1. Introduction

Current guidelines provide strong recommendations for oral anticoagulation as the first-line therapy for stroke prevention in patients with non-valvular atrial fibrillation (AF) [1,2]. However, specific patient subsets are not adequately anticoagulated due to contraindications or low adherence to anticoagulation [3], typically older and at risk for bleeding or ischemic complications [4]. Therefore, based on longer-term safety and effectiveness data, percutaneous occlusion has been recommended as an alternative to anticoagulation for these patients [5,6]. Although concomitant surgical left atrial appendage (LAA) intervention at the time of surgery seems likely to be superior concerning the anticipated risk of the subsequent percutaneous procedure, only a tepid recommendation for surgical LAA occlusion at the time of cardiac surgery has been provided for patients with AF, which is primarily due to the lack of high-quality evidence [1,2].

The Left Atrial Appendage Occlusion Study (LAAOS III) recently provided new and compelling data to guide clinical decisions regarding surgical occlusion as an adjunct procedure to main primary cardiac surgery in patients with AF [7]. However, the risk of ischemic stroke might not be uniform across subpopulations with AF undergoing cardiac surgery, whereby patients with aortic stenosis or after biological aortic valve replacement (AVR) were found at increased thromboembolic risk [8,9,10]. Furthermore, different LAA closing techniques have demonstrated variable effectiveness regarding completeness of occlusion, the lowest for running, purse-string sutures or external LAA ligation [11,12].

Therefore, the present study aimed to assess thromboembolic risk after LAA amputation in patients with leading aortic stenosis, pre-existing AF undergoing biologic AVR with or without concomitant myocardial revascularization.

## 2. Patients and Methods

Initially, 602 patients with concomitant AF scheduled for first-time aortic valve surgery and/or coronary artery bypass grafting (CABG) at the Department of Cardiac Surgery, Klinikum Nuerberg, Paracelsus Medical University, Nuernberg from January 2013 to January 2019 were identified from archived patient files. A minimum of 12 months since the index surgery was required for a patient to be eligible. Due to admittedly differing thromboembolic risk, patients scheduled for isolated CABG, cardiac reoperation, mitral valve surgery (typically undergoing an endoscopic procedure, AF ablation and alternative LAA exclusion), and patients with a history of pulmonary embolism or deep vein thrombosis [13] were not considered for the analysis. Patients having received a mechanical valve prosthesis were excluded because of having a potential additional source of embolism and an absolute indication for continued anticoagulation. Patients undergoing AVR with biological prosthesis and/or CABG were divided into two groups based on amputation of LAA. The amputation of LAA was performed either with amputation followed by direct oversewing suture or by a surgical stapler (Covidien, Medtronic, Dublin, Ireland). If a patient had a thrombus in the LAA, the left atrium was opened to remove the thrombus before occlusion. Purse-string, double-layer running suture or external LAA ligation were not permitted, and closure with an approved epicardial surgical occlusion device was not contemplated for LAA amputation in this study. Patient’s characteristics, risk factors, surgical details, and outcome data were retrieved from SAP (Waldorf, Germany) and THG-QIMS (Terraconnect, Nottuln, Germany) hospital quality management software. Prescribed medication was meticulously recorded along with the calculation of CHA_2_DS_2_-VASc and HAS-BLED scores, whereby the status of LAA (non) amputation did not serve as criterion for suspension of anticoagulant therapy at any point. Therewith, 157 patients, 74 with concomitant vs. 83 without LAA amputation, were included for further analyses. Moreover, to mitigate the effects of measurable cofounders, patients were matched into 74 pairs according to propensity score matching based on preoperative characteristics. The flowchart displaying the patient data and activity flow is presented in Figure 1.

### 2.1. Ethical Statement

The study was conducted in accordance with the Declaration of Helsinki and approved by the Institutional Review Board on 24 January 2020 (IRB-2020-006). All patients signed an agreement at the time of admission to use their data and future contact permit for follow-up, control, analysis and publication of anonymized data. Formal informed consent was waived due to the retrospective design, utilizing routinely obtained de-identified clinical and laboratory data.

### 2.2. Definitions

Ischemic stroke included transient ischemic attack with positive neuroimaging [14] and any stroke excluding definite hemorrhagic stroke. Severe stroke has been associated with neurological residua impacting the daily activities and defined by modified Rankin Scale ≥ 2 [15]. A major bleeding event was defined as type 2 or 3 bleeding requiring hospitalization [16].

### 2.3. Follow-Up

Participants were followed up by telephone (personally or through interview of an aligned general practitioner) or in person. Particular attention during follow-up was given to collecting data on cerebrovascular events, including stroke and transient ischemic attack (TIA). The questions used in the follow-up phone interview were on current medication, possible stroke/TIA from the operation to the call, heart rate and rhythm, possible anticoagulation-related events, bleeding, any other operations.

### 2.4. Statistical Analysis

Categorical data were reported as frequencies and proportions, and the differences between groups were tested with the Chi-square test with continuity correction and *p*-values and odds ratios (OR) with 95% confidence intervals (95% CI) were reported. Continuous data were summarized as the mean (standard deviation) if normally distributed and as median (1st quartile; 3rd quartile) in non-normal cases. The differences between groups were tested with Mann–Whitney U-tests and considered significant when *p* < 0.05. Follow-up analysis of patients was made by using Kaplan–Meier curves, where differences between LAA vs. No LAA amputation group were compared using the log-rank test. Further, the Cox regression analysis was performed, including the variables LAA amputation, concomitant AF ablation, CHA_2_DS_2_-VASc score, history of any stroke, prescribed antiplatelet agents vitamin K-antagonists, direct oral anticoagulants and presence of AF on ECG at time of discharge. The results of the Cox regression modeling were reported in terms of hazard ratios with 95% confidence intervals and with significances of each variable in the model.

## 3. Results

The final cohort included 157 patients. The preoperative clinical profile of 74 patients with concomitant and 83 without LAA amputation is summarized in Table 1. Patients with concomitant LAA amputation were younger (74 (69;77) vs. 77 (73;79) years; *p* = 0.012), had lower operative risk assessed by EuroSCORE (8.1 (4.7;16.5) vs. 10.3 (8.1;15.3); *p* = 0.044), tendency toward higher male predominance (56 (75.7%) vs. 51(61.4%); *p* = 0.082) as well as insignificantly better preserved left ventricular ejection fraction (55 (46;60) vs. 60 (48;64); *p* = 0.062. Therefore, to mitigate the potential confounding effects, propensity score matching was performed to yield 74 pairs with similar baseline characteristics as presented in the right columns of the Table 1. Regardless of the matching status, no differences were found with regard to history of stroke, AF type, stroke and bleeding risks as defined by CHA_2_DS_2_-VASc and HAS-BLED scores and prescription of oral anticoagulants between LAA and No-LAA amputation groups (Table 1).

Nonetheless, patients with concomitant LAA amputation had undergone more complex surgery, receiving isolated AVR through upper mini sternotomy less frequently (*p* < 0.001), but more often concomitant revascularization and AF ablation through full sternotomy (*p* < 0.001) in both preoperatively unmatched and matched population, as depicted in Table 2. Increased complexity was reflected in significantly longer cross-clamp and cardiopulmonary bypass times in the LAA amputation group in unmatched as well as the matched cohort (*p* < 0.001 for both) (Table 2). Significantly fewer patients in the LAA amputation group received stentless aortic bioprosthesis (*p* = 0.008 and *p* = 0.018 for the unmatched and matched cohort, respectively). LAA excision was performed by cutting and sewing in 32 and utilizing staples in 31 patients from the LAA amputation group. Additional hemostatic Teflon-pledgeted suture was placed after excision in 11 of 32 patients after LAA amputation by stapler. Similar late reoperation rates for pericardial effusion in the same hospitalization were found in LAA vs. No-LAA amputation group in both the unmatched and matched cohort (*p* = 1.000 for both), respectively (Table 2).

Postoperative ischemic stroke and 30-day mortality were comparable in unmatched (*p* = 0.448 and *p* = 0.685, respectively) and remained so after propensity matching (*p* = 1.000 and *p* = 0.681), as presented in Table 2. The cause of death could be attributed to ischemic stroke in one patient from the No-LAA amputation group (Table 2). Patients with and without LAA amputation were discharged from the hospital with similar proportions of AF (55.4 vs. 55.4%, *p* = 1.000) and anticoagulants (94.4% and 90.1%; *p* = 1.000); the proportions remained unchanged after propensity matching (Table 2).

From 151 survivors of the original unmatched cohort, three in the LAA amputation group (4.2%) and two in the No-LAA amputation group (2.5%) refused further participation, and seven patients from each group had been lost to follow-up (9.3%). Thus, final follow-up for the primary outcome of ischemic stroke or death was performed in 137 patients (90.7%) and completed for all clinical variables in 132 patients (87.4%) with a median follow-up of 48 (29;66) vs. 46 (31;67) months in the LAA amputation vs. No-LAA amputation group (*p* = 0.787; Table 3). Late ischemic stroke occurred in two patients in the LAA amputation group (3.2%) and 12 (17.4%) patients in the No-LAA group (*p* = 0.018; Table 3). One patient (1.6%) in the LAA group and four (5.8%) in the No-LAA amputation group suffered from severe stroke (*p* = 0.445). Mortality and hospitalization rates, specifically cardiovascular related, were comparable between patients that received LAA amputation and those that did not (Table 3).

After matching, all the primary and secondary outcomes remained unchanged. In particular, cumulative (6.4% vs. 25%, *p* = 0.028), late ischemic (3.2% vs. 20%, *p* = 0.008) and any stroke (7.9% vs. 26.6%, *p* = 0.037) occurred more frequently in the No-LAA amputation cohort (right panel of Table 3). A trend toward increase hospitalizations for any cause was observed in the No-LAA cohort (23.8% vs. 45%, *p* = 0.085, Table 3).

Cox–hazard analysis identified LAA amputation as the only significant factor, reducing the incidence of ischemic stroke in unmatched (hazard ratio, 0.26; 95% confidence interval (CI) 0.09–0.79; *p* = 0.017) and matched (hazard ratio, 0.30; 95% confidence interval (CI) 0.11–0.84; *p* = 0.021, Figure 2 and Figure 3). No other peripheral systemic embolization was recorded during the follow-up in any patient.

## 4. Comment

Among the patients with AF and leading aortic stenosis undergoing biological AVR with or without concomitant myocardial revascularization, the risk of ischemic stroke was reduced with concomitant LAA amputation during cardiac surgery.

In a retrospective study of Elbadawi et al. including 1304 patients with AF undergoing valvular surgery, fewer postoperative strokes were reported (2.5% vs. 4.6%) in the LAA exclusion group with CHA_2_DS_2_-VASc score of ≥2), whereby significantly higher rates of bleeding, pericardial tamponade and higher in-hospital mortality rates were observed. Concomitant surgical ablation did not demonstrate additional benefit on the primary outcome of postoperative stroke (2.1% vs. 2.6%, *p* = 0.73) [17]. Similarly, 2.2% vs. 2.7% ischemic stroke and/or systemic embolism rates were reported in a mixed population undergoing cardiac surgery with LAA occlusion vs. No-LAA occlusion, respectively, in the LAAOS III study [7], without any additional benefit of concomitant surgical ablation. A higher incidence of ischemic stroke regardless of LAA intervention in our study is in line with the findings of Andreasen et al. [10], identifying aortic stenosis and AF as an exceptionally high-risk combination for thromboembolisms. Assumingly, mixed patient cohorts with diverse LAA occlusion modalities [7,8,18] arguably precluded large-scale analyses from demonstrating more significant benefit of LAA intervention in patients with concomitant aortic stenosis/AVR and AF. Although a longer cardiopulmonary bypass is a known risk factor for stroke [9], we observed a trend toward higher perioperative stroke (6% vs. 2.7%, *p* = 0.448) in the no-LAA amputation group despite shorter CPB. There were comparable rates of reoperations for bleeding, tamponade or pericardial effusion.

The observed overall stroke reduction in patients with LAA amputation is consistent with the results of preceding studies [7,8]. The reported benefit of stroke and systemic embolism reduction by Whitlock et al. (4.8% vs. 7.0%, adjusted HR 0.67; *p* = 0.001) and Friedman et al. (4.2% vs. 6.2%, adjusted HR 0.26; *p* < 0.001) during a comparable mean follow-up of 3.8 and 2.6 years, respectively [7,8] seem to lie within a similar range. Notably, the anticipated benefits of LAA occlusion were associated with a lower risk of thromboembolism only among patients without anticoagulation at discharge by Friedman et al. [8] as opposed to over 80% of anticoagulated patients at discharge in the study of Whitlock et al. [7]. With anticoagulation at discharge exceeding 90%, we observed a higher risk of stroke in patients with prior AF not receiving LAA amputation at the time of AVR. Wilbring et al. reported comparable stroke rates in a registry cohort of 398 patients with permanent AF undergoing any cardiac surgery [19]. Of note, similar stroke incidence and sinus rhythm rates at 1-year follow-up in patients with LAA closure alone or in combination with surgical ablation as opposed to exceptionally high stroke incidence in patients without LAA intervention (7.1% vs. 6.5 vs. 20.5%, *p* < 0.01) implied no or little additional impact of ablation on stroke rate reduction [19]. Similarly, surgical ablation did not translate into an additive preventive effect on our cohort’s stroke rate.

Substantial hemodynamic and neurohormonal changes affecting the RAAS and ANS system were reported following percutaneous epicardial LAA exclusion [20], possibly potentiating heart, renal and respiratory failure as well as recurrences of AF also in patients with concomitant AF after coronary artery bypass grafting [21]. Several other mixed population studies reported more frequent AF episodes and failed to demonstrate stroke reduction in patients without prior AF undergoing concomitant LAA exclusion, however with no account to the exclusion technique used [22,23]. Likewise, Gutierrez et al. reported beneficial effects of LAA closure only in patients with pre-existing AF [18]. Similarly to Whitlock et al. [7], no increase in (re)hospitalization rate due to AF recurrences or heart failure was observed in our arguably different cohort of patients with poststenotically altered but decompressed myocardium, preserved left ventricular ejection fraction and similar proportions of AF on ECG at discharge.

Thus, threefold stroke reduction after LAA amputation in our cohort with concomitant AF and leading aortic stenosis regardless of CHA2DS2-VASc score identifies patients potentially benefiting from LAA occlusion the most. At present, typically only 20% of patients with concomitant AF received LAA closure at the time of AVR [24], possibly reflecting the fact that nowadays, many centers operate on the aortic valve using a minimally invasive approach with peripheral cannulation whereby the LAA amputation in this setting might not be the easiest maneuver to do. As less invasive approaches are increasingly used for surgical AVR with appealing outcomes [25], we estimate that life-saving LAA interventions could be performed more often by using an epicardial device in order to overcome the issues of limited accessibility.

## 5. Study Limitations

The present study was a single-center, retrospective database analysis. The exact status of anticoagulation at discharge but not at the time of follow-up could be reliably inferred. The follow-up information could be performed in person for one-third of patients, for one-quarter of patients from a general practitioner and from the rest by telephone interview only. However, it has also the following strengths: the consistently used method of LAA occlusion was uniformly an amputation, minimizing the possible ensuing complications from incomplete LAA occlusion. The investigated cohort including patients with leading aortic stenosis was more homogeneous with regard to comparable studies reporting the results in mixed populations.

## 6. Conclusions

Patients with AF and planed AVR have lower incidence of CVI associated with LAA amputation. LAA amputation proved to be safe. Given the unique opportunity of addressing LAA to reduce ischemic stroke, surgical strategies, including novel LAA occlusion modalities, should be contemplated more often in high-risk patients with stenotic aortic valve and concomitant AF.

## Figures and Tables

**Figure 1 jcm-11-03408-f001:**
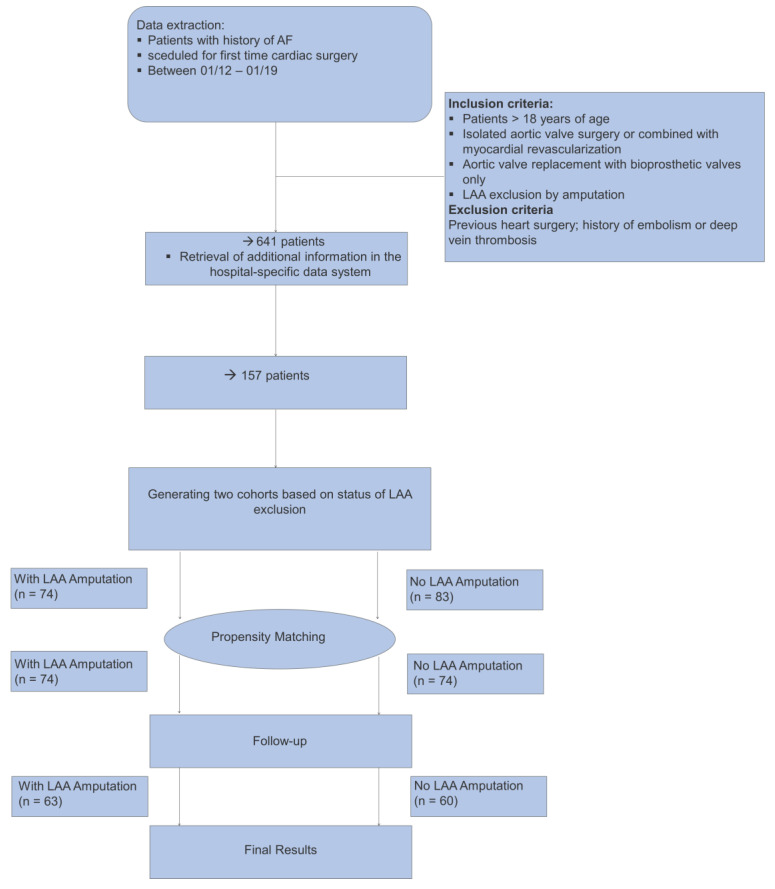
Flowchart displaying the patient data and activity flow.

**Figure 2 jcm-11-03408-f002:**
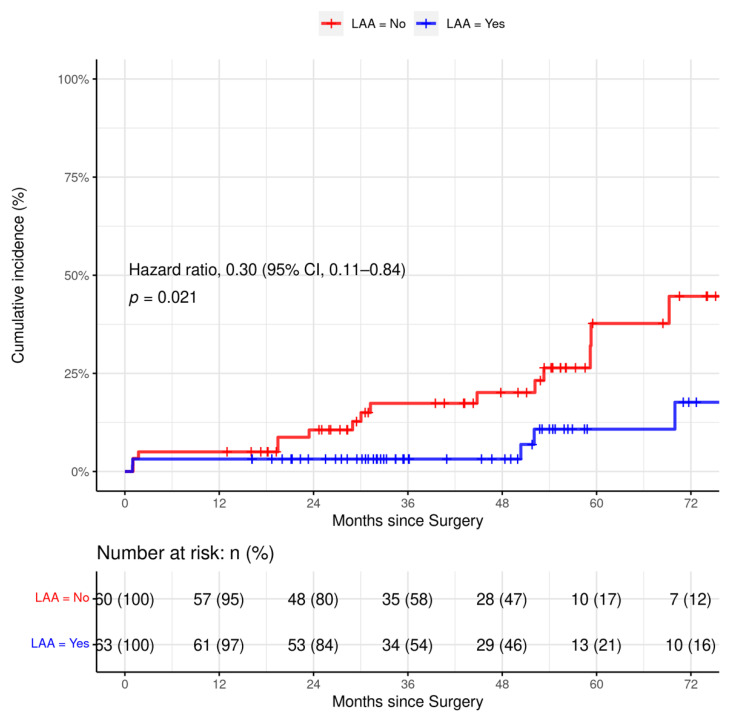
Incidence of cumulative ischemic stroke in matched patients with left atrial appendage vs. without left atrial appendage amputation.

**Figure 3 jcm-11-03408-f003:**
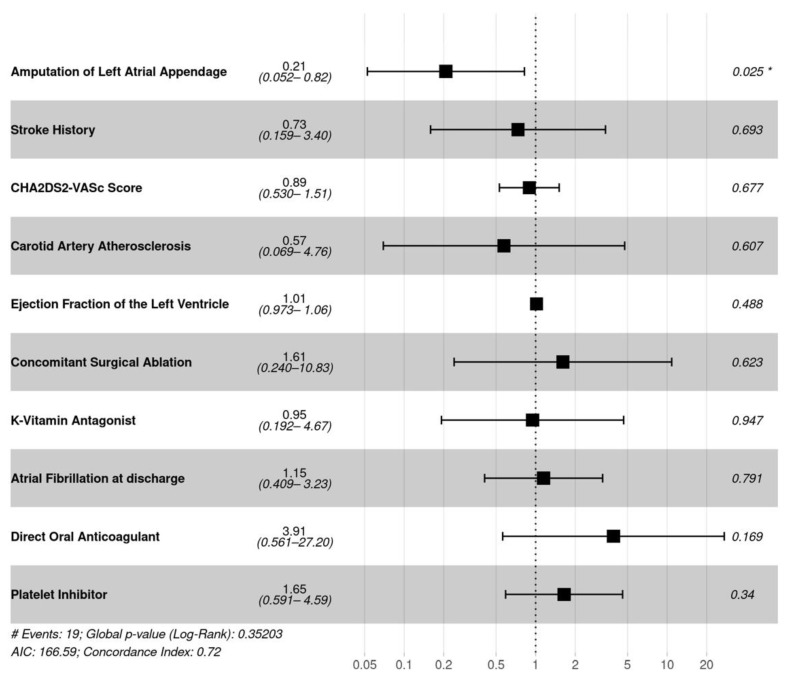
Analysis of the effect of potential covariates on ischemic stroke in matched patients with versus without left atrial appendage amputation. * significant hazard ratio.

**Table 1 jcm-11-03408-t001:** Preoperative profile of 157 unmatched and 148 matched patients with LAA vs. No-LAA Amputation.

	Unadjusted Data			Propensity Score Matched Data		
Variable	LAA Amputation *n* = 74	No-LAA Amputation *n* = 83	*p*-Value	LAA Amputation *n* = 74	No-LAA Amputation *n* = 74	*p*-Value
Age (years) *	74 (69;77)	77 (73;79)	0.012	74.0 (69.0;77.0)	76.0 (73.0;78.0)	0.069
BMI (kg/m^2^)	28.4 (25.6;31.9)	27.3 (24.7;31.2)	0.252	28.4 (25.6;31.9)	27.6 (24.7;31.2)	0.333
Carotid artery disease (%)	10 (13.5%)	8 (9.6%)	0.610	10 (13.5%)	8 (10.8%)	0.801
Coronary artery disease (%)	46 (62.2%)	41 (49.4%)	0.148	46 (62.2%)	37(50.0%)	0.185
CHA2DS2-VASc score *	4 (4;5)	4 (4;5)	0.313	4 (4;5)	4 (4;5)	0.534
Chronic kidney disease (%)	3 (4.1%)	6 (7.2%)	0.502	3 (4.1%)	6 (8.1%)	0.494
Diabetes Mellitus II (%)	30 (40.5%)	29 (34.9%)	0.577	30 (40.5%)	27 (36.5%)	0.735
Dyslipidemia	61 (82.4%)	69 (83.1%)	1.000	61 (82.4%)	63 (85.1%)	0.824
EuroScore I *	8.1 (4.7;16.5)	10.3 (8.1;15.3)	0.044	8.1 (4.7;16.5)	10.0 (7.2;14.7)	0.087
HAS-BLED Score	2 (2;3)	3 (2;3)	0.195	2 (2;3)	3 (2;3)	0.172
History of heart failure (%)	9 (12.2%)	10 (12.0%)	1.000	9 (12.2)	10 (13.5%)	1.000
History of ischemic stroke	12 (16.2)	15 (18.1)	0.924	12 (16.2)	13 (17.6%)	1.000
Hypertension	72 (97.3)	82 (98.8)	0.602	72 (97.3)	73 (98.6%)	1.000
LVEF *	55 (46;60)	60 (48;64)	0.062	55 (46;60)	60 (45;60)	0.256
Male gender (%)	56 (75.7%)	51(61.4%)	0.082	56 (75.7%)	51 (68.9%)	0.463
MI within 3 weeks (%)	4 (5.4%)	2 (2.4%)	0.422	4 (5.4%)	2 (2.70%)	0.681
Paroxysmal AF (%)	26 (35.1%)	40 (48.2%)	0.136	26 (35.1%)	36 (48.6%)	0.134
Persistent AF (%)	22 (29.7%)	15 (18.1%)	0.126	22 (29.7%)	13 (17.6%)	0.122
Permanent AF (%)	26 (35.1%)	28 (33.7%)	0.987	26 (35.1%)	25 (33.8%)	1.000
Peripheral arterial disease (%)	3 (4.1%)	6 (7.2%)	0.502	3 (4.1%)	6 (8.1%)	0.494
Preoperative creatinine (mg/dL) *	1.1 (1;1.4)	1.1 (0.9;1.3)	0.322	1.1 (1;1.4)	1.1 (0.9;1.3)	0.472
Therapy before surgery						
Vitamin K antagonist (%)	26 (35.1%)	28 (33.7%)	0.987	26 (35.1%)	25 (33.8%)	1.000
Direct oral anticoagulant (%)	24 (32.4%)	27 (32.5%)	1.000	24 (32.4%)	23 (31.1%)	1.000
Platelet Inhibitor (%)	21 (28.4%)	25 (30.1)	0.949	21 (28.4%)	23 (31.1%)	0.857

* (Q1;Q3) = median (1st quartile;3rd quartile). AF = atrial fibrillation; BMI = body mass index; LAA = left atrial appendage; LVEF = left ventricular ejection fraction; MI = myocardial infarction.

**Table 2 jcm-11-03408-t002:** Operative characteristics of patients with LAA vs. No-LAA amputation, unmatched and matched according to baseline characteristics.

	Unadjusted Data			Propensity Score Matched Data		
Variable	LAA Amputation *n* = 74	No-LAA Amputation *n* = 83	*p*-Value	LAA Amputation *n* = 74	No-LAA Amputation *n* = 74	*p*-Value
Upper partial sternotomy (%)	17 (23)	59 (71.1)	<0.001	17 (23)	55 (74.3)	<0.001
Isolated aortic valve replacement (%)	33 (44.6)	65 (78.3)	<0.001	33 (44.6)	60 (81.1)	<0.001
Concomitant revascularization (%)	41 (55.4)	18 (21.7)	<0.001	41 (55.4)	14 (18.9)	<0.001
Concomitant surgical ablation of AF (%)	26 (35.1)	1 (1.2)	<0.001	26 (35.1)	0 (0.0)	<0.001
Cardiopulmonary bypass *, min	103 (81;126)	70 (56;97)	<0.001	103 (81;126)	71 (56;98)	<0.001
Aortic cross-clamping time *, min	70 (54;88)	44 (32;66)	<0.001	70 (54;88)	49 (32;67)	<0.001
Sutureless biological prosthesis (%)	21 (28.4)	42 (50.6)	0.008	21 (28.4)	36 (48.6)	0.018
Stapler/Cut and sew for LAA amputation	31/32	NA		31/32	NA	
Revision for bleeding/tamponade in 48 h (%)	3 (4.1)	6 (7.2)	0.502	3 (4.1)	6 (8.1)	0.494
Red blood cell transfusion *, units	1 (1;1)	1 (1;2)	0.230	1 (1;1)	1 (1;1)	0.551
Late operation for pericardial effusion (%)	2 (2.7)	3 (3.6)	1.000	2 (2.7)	3 (4.1)	1.000
New pacemaker due to AV block (%)	3 (4.1)	3 (3.6)	1.000	3 (4.1)	3 (4.1)	1.000
ICU Stay *, (d)	2 (1;5)	2 (1;5)	0.734	2 (1;5)	2 (1;5)	0.959
Hospital Stay *, (d)	12 (8;15)	12 (9;16)	0.610	12 (8;15)	12 (9;16)	0.660
AF on ECG at discharge (%)	41 (55.4)	46 (55.4)	1.000	41 (55.4)	41 (55.4)	1.000
Therapy at discharge						
Vitamin K antagonist (%)	63 (87.5)	67 (83.8)	0.671	63 (87.5)	61 (85.9)	0.974
Direct oral anticoagulant (%)	5 (6.9)	5 (6.3)	1.000	5 (6.9)	4 (5.6)	1.000
Platelet Inhibitor (%)	46 (62.2)	38 (46.3)	0.069	46 (62.2)	34 (46.6)	0.083
Ischemic stroke within 30 days (%)	2 (2.7)	5 (6)	0.448	2 (2.7)	3 (4.1)	1.000
Mortality within 30 days (%)	2 (2.7)	4 (4.8)	0.685	2 (2.7)	4 (5.4)	0.681

* median (Q1;Q3) = (1st quartile;3rd quartile) AV = atrioventricular; d = day; ECG = electrocardiogram; ICU = intensive care unit; LAA = left atrial appendage.

**Table 3 jcm-11-03408-t003:** Outcomes of patients with LAA vs. No-LAA amputation at follow-up, unmatched and matched with respect to baseline.

	Unadjusted Data			Propensity Score Matched Data		
Variable	LAA Amputation *n* = 63	No-LAA Amputation *n* = 69	*p*-Value	LAA Amputation *n* = 63	No-LAA Amputation *n* = 60	*p*-Value
Follow-up; median * (months)	48 (29;66)	46 (31;67)	0.787	48 (29;66)	45 (27;64)	0.494
Primary						
Cumulative ischemic stroke (%)	4 (6.4)	17 (24.6)	0.026	4 (6.4)	15 (25.0)	0.028
Secondary						
Late ischemic stroke beyond 30 days (%)	2 (3.2)	12 (17.4)	0.018	2 (3.2)	12 (20.0)	0.008
Any stroke	5 (7.9)	17 (24.6)	0.019	5 (7.9)	16 (26.6)	0.037
Fatal ischemic stroke (%)	1 (1.6)	1 (1.5)	1.000	1 (1.6)	1 (1.6)	1.000
Severe ischemic stroke (Rankin score > 2; %)	1 (1.6)	4 (5.8)	0.445	1 (1.6)	4 (6.6)	0.361
Fatal hemorrhagic stroke	1 (1.6)	0 (0.0)	0.970	1 (1.6)	0 (0.0)	1.000
Major bleeding (%)	2 (3.2)	2 (2.9)	1.000	2 (3.2)	2 (3.3)	1.000
Systemic embolism	0	0	NA	0	0	NA
Hospitalizations for any cause (%)	15 (23.8)	29 (42)	0.161	15 (23.8)	27 (45.0)	0.085
Hospitalization for cardiovascular cause (%)	10 (15.9)	9 (13)	0.877	10 (15.9)	7 (11.7)	0.985
Death from any cause (%)	15 (23.8)	12 (17.4)	0.486	15 (23.8)	9 (15.0)	0.315
Cardiovascular + unexplained death (%)	8 (12.7)	4 (5.8)	0.340	8 (12.7)	7 (11.7)	0.472
Non-cardiovascular death (%)	7 (11.1)	8 (11.6)	1.000	7 (11.1)	5 (8.3)	0.830

* median (Q1;Q3) = (1st quartile;3rd quartile) LAA = left atrial appendage; NA = not applicable.

## Data Availability

The data presented in this study are available on request from the corresponding author. The data are primarily not publicly available due to the data protection policy of the institution.

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
