# Peer review of "Left Atrial Appendage Amputation for Atrial Fibrillation during Aortic Valve Replacement"

_jcm, 2022, doi:10.3390/jcm11123408_

Round 1

Reviewer 1 Report

Dear Dr. Kalisnik and co-authors. Congratulations on well written paper.

I have several issues/ comments.

  1. In the Abstract Background  you are stating that " Whether occluding the LAA during open heart surgery reduces the risk of ischemic stroke is debated but later in the body of the manuscript you mention good quality studies which show clear benefit. So it's not debated anymore, or at least should not be.
  2. Two cohorts with LAA amputation and No-LAA amputation are not perfectly matched therefore comparing of the outcomes in these groups are somewhat limited. I would recommend a propensity score matching since the retrospective nature of this study.
  3. In figure 3 there is total follow up is 303 and 333 months. What does it mean?
  4. In Figure 4 quality of life parameters are consistently better in LAA-A group, how can you explain this? does it change after propensity score matching?

Thank you very much.

Reviewer 2 Report

Very interesting paper with very good results on LAA amputation, supported and supporting the literature. Very well conducted. I would suggest to insert in the discussion one issue: nowadays many center operate on the aortic valve in minimally invasive, with peripheral cannulation. In this setting the amputation is not the easiest maneuver, but using an epicardial device can be the solution. I would elaborate on it, amputation and proper suturing of LAA is much more safe in full sternotomy.

Reviewer 3 Report

Dear Authors

This is an interesting study that addresses an important problem of amputation of the the ‘clinical efficacy’ of left atrial appendage during aortic valve replacement  in patients with atrial fibrillation.

This is a retrospective study based on clinical records including patients operated upon between 2013 to 2019  and a follow up   of median  duration of 48 or 46 months.

Those findings seem to me very interesting and can impact clinical practice.

My main critical remarks are related to information in the follow-up phase ;

  1. Provide more data on the methods of the follow-up. It was stated that it was  a telephone interview or in person. How many are seen in person ?
  2. What kind of question were asked  in order to differentiate stroke type, cardiovascular vs non-cardiovascular death or major bleeding (please provide used definitions and an explanation how those problems were addressed in the telephonic interview?
  3.  How was the diagnosis of non-hemorrhagic stroke during follow-up? This information was positive only when received by telephone conversation or a additional documentation or visit in person or documentation analyzed ?
  4.  Provide information whether those patients were also on the quality of antithrombotic therapy – most of the patients were discharge d on vitamin K antagonists. If not, please add this information as an important limitation of the study.
  5. Which period of time cover figure 2 KM curves. In hospital and follow-up period or only after discharge ?  -Please specify it in the explanation in Fig. 2
